# Different Effects of Cyclical Ketogenic vs. Nutritionally Balanced Reduction Diet on Serum Concentrations of Myokines in Healthy Young Males Undergoing Combined Resistance/Aerobic Training

**DOI:** 10.3390/nu15071720

**Published:** 2023-03-31

**Authors:** Pavel Kysel, Denisa Haluzíková, Iveta Pleyerová, Kateřina Řezníčková, Ivana Laňková, Zdeňka Lacinová, Tereza Havrlantová, Miloš Mráz, Barbora Judita Kasperová, Viktorie Kovářová, Lenka Thieme, Jaroslava Trnovská, Petr Svoboda, Soňa Štemberková Hubáčková, Zdeněk Vilikus, Martin Haluzík

**Affiliations:** 1Department of Sports Medicine, First Faculty of Medicine and General University Hospital, 128 08 Prague, Czech Republic; 2Centre for Experimental Medicine, Institute for Clinical and Experimental Medicine, 140 21 Prague, Czech Republic; 3Diabetes Centre, Institute for Clinical and Experimental Medicine, 140 21 Prague, Czech Republic; 4Institute of Medical Biochemistry and Laboratory Diagnostics, First Faculty of Medicine, Charles University and General University Hospital, 121 11 Prague, Czech Republic; 5Department of Biochemistry and Microbiology, University of Chemistry and Technology Prague, Technicka 5, 166 28 Prague, Czech Republic

**Keywords:** body composition, ketogenic diet, strength parameters, endurance, training, myokines, adipokines, cytokines

## Abstract

Myokines represent important regulators of muscle metabolism. Our study aimed to explore the effects of a cyclical ketogenic reduction diet (CKD) vs. a nutritionally balanced reduction diet (RD) combined with regular resistance/aerobic training in healthy young males on serum concentrations of myokines and their potential role in changes in physical fitness. Twenty-five subjects undergoing regular resistance/aerobic training were randomized to the CKD (*n* = 13) or RD (*n* = 12) groups. Anthropometric and spiroergometric parameters, muscle strength, biochemical parameters, and serum concentrations of myokines and cytokines were assessed at baseline and after 8 weeks of intervention. Both diets reduced body weight, body fat, and BMI. Muscle strength and endurance performance were improved only by RD. Increased musclin (32.9 pg/mL vs. 74.5 pg/mL, *p* = 0.028) and decreased osteonectin levels (562 pg/mL vs. 511 pg/mL, *p* = 0.023) were observed in RD but not in the CKD group. In contrast, decreased levels of FGF21 (181 pg/mL vs. 86.4 pg/mL, *p* = 0.003) were found in the CKD group only. Other tested myokines and cytokines were not significantly affected by the intervention. Our data suggest that changes in systemic osteonectin and musclin levels could contribute to improved muscle strength and endurance performance and partially explain the differential effects of CKD and RD on physical fitness.

## 1. Introduction

A diet with a nutritionally suitable composition is a necessary prerequisite for both professional and amateur sportsmen to enable appropriate energy expenditure and ensure optimal physical performance. For aesthetic and performance reasons, these subjects frequently undergo weight loss in order to reduce body fat amount. However, this may be accompanied by a loss of lean body mass, which can often reach up to 25% of the total weight loss [1,2]. For this purpose, the aim of such approaches is to maximize the reduction of adipose tissue while maintaining muscle mass. To achieve weight loss, it is necessary to increase the energy deficit, which most often means reducing energy intake. There are many types of reduction diets; however, most of them are similar in many points [3]. A caloric reduction of 500–750 calories per day is recommended by many obesity societies [4,5]. To reduce the number of calories, lipids or carbohydrates are usually restricted. Neither macronutrient, fat (low-fat diets) nor carbohydrate (low-carb diets), has been determined to be more important for weight loss as long as a caloric deficit occurs. An extremely low energy intake from carbohydrates (<10%) may result in nutritional ketosis [6]. This type of diet is called the ketogenic diet and is usually accompanied by elevated fat and protein content, with a daily protein intake of 0.8–1.5 g/kg body weight to preserve muscle mass [6,7]. The magnitude and duration of the energy deficit determine the amount of weight loss. Resistance exercise is often used to restrict the loss of lean mass [8].

Skeletal muscle and bone are connected anatomically and physiologically and play a crucial role in human locomotion and metabolism. Historically, the coupling between muscle and bone has been viewed in the light of mechanotransduction, which declares that the mechanical forces applied to muscle are transmitted to the skeleton to initiate bone formation. However, these tissues also communicate through an endocrine route orchestrated by a family of bioactive molecules referred to as myokines (derived from myocytes), osteokines (derived from bone cells), and adipokines (derived from adipose tissue) [9].

Myokines are small proteins, released by skeletal muscle cells in response to muscle contractions, which have important autocrine, paracrine, and/or endocrine effects, including the regulation of energy metabolism. Experimental studies have demonstrated that some of the myokines can be directly involved in the regulation of muscle strength and endurance, thus possibly contributing to the integration of the effects of different diets and various types of exercise on the parameters of physical fitness [10].

One of the myokines with important effects on metabolic regulation and muscle growth and performance is musclin. Musclin is produced by bone and skeletal muscle cells and plays a crucial role in the regulation of bone growth and physical endurance by enhancing the growth of muscle fibers and their regeneration [11,12,13]. In experimental settings, musclin attenuated inflammation and oxidative stress in doxorubicin (DOX)-induced cardiotoxicity [14].

Osteonectin, also known as secreted protein acidic and rich in cysteine (SPARC), is another important tissue-specific protein, linking the bone mineral and collagen phases and contributing to the initiation of active mineralization in normal skeletal tissue. Osteonectin is an important modulator of the actin cytoskeleton that may be involved in the maintenance of muscular function by binding to actin in regenerating myofibers [15,16,17]. Several other myokines, such as interleukin 6 (IL6), irisin, insulin growth factor (IGF-1), brain-derived neurotrophic factor (BDNF), myostatin, and fibroblast growth factor 21 (FGF21), exert anabolic/catabolic effects on bone and muscle, and their appropriate balance contributes to the physiological regulation of muscle and bone mass [9]. Especially FGF21 was found to be required for fasting-induced muscle atrophy and weakness [18].

In addition to its effects on body composition, there is growing evidence that a ketogenic diet may be a potential therapeutic approach for Alzheimer’s disease or other types of cognitive impairment via inhibition of pro-inflammatory cytokines, such as interleukin-1β (IL-1β) and tumor necrosis factor α (TNF-α) [19,20,21]. These cytokines are also known to accelerate catabolism, induce contractile dysfunction, disrupt myogenesis, and overall modulate muscle tissue loss (summarized in [22]). Their inhibition can therefore lead to improved muscle function. Moreover, a ketogenic diet may improve insulin sensitivity by reducing carbohydrate intake and promoting weight loss. This can help prevent the development of type 2 diabetes and also, in turn, reduce the risk of sleep apnea and affect a wide variety of other physiological regulations, including the endocannabinoid system [23,24,25,26].

Taken together, myokines represent an important interconnection between the effects of diet and exercise and the growth of muscle fibers and their regeneration, modulation of angiogenesis, and also the regulation of glucose and lipid metabolism [27]. We have previously demonstrated that a nutritionally balanced reduction diet (RD), but not cyclical ketogenic diet (CKD), in healthy young males undergoing regular resistance/aerobic training improved muscle strength and endurance performance, while both reduced body weight to a similar degree [28].

Here, we explored possible mechanisms behind the different effects of RD vs. CKD on physical performance. The main aim of this study was to test the hypothesis that exercise-induced myokines are produced differently according to the type of diet protocol selected and thus differentially affect physical performance. To this end, we measured serum concentrations of selected myokines and cytokines in healthy young males on RD or CKD undergoing regular resistance/aerobic training at baseline and after eight weeks of training combined with their respective diets.

## 2. Materials and Methods

### 2.1. Study Subjects

In our earlier paper [28], the specific study subjects’ characteristics and the training regimen were laid forth in detail. In summary, 25 healthy young male volunteers with a wide range of fitness and at least a year of resistance training experience were recruited through physical education colleges and through a website with people interested in nutrition and exercise. Ages between 18 and 30 years and at least a year of resistance and aerobic training experience were requirements for inclusion. Participants who expressed interest in taking part were assessed to make sure they met the requirements for enrollment in this study.

Exclusion criteria included the existence of cardiovascular disorders, diabetes mellitus, arterial hypertension, or any other illnesses that required pharmacological treatment. These ailments could impact athletes’ ability to perform in sports or put them at risk for subsequent injuries. Additionally, participants were required to stop using any performance-enhancing supplements (such as creatine, β-hydroxy-β-methyl butyrate, caffeine, protein powder, weight gainer, thermogenic, etc.) at least one week before baseline testing and to refrain from doing so for the duration of this study.

This study was authorized by the Human Ethics Review Board, First Faculty of Medicine and General University Hospital, Prague, Czech Republic (ethical approval code 764/18 S-IV) and performed in agreement with the principles of the WMA Declaration of Helsinki—Ethical Principles for Medical Research Involving Human Subjects [29]. All subjects were required to sign informed consent forms before randomization.

For eight weeks, participants were randomized by an electronic randomization system to follow either CKD or RD (both requiring a 500 kcal/day reduction in total calorie intake) while engaging in three resistance exercise sessions and three cardio exercises per week (30 min run, heart rate around 130–140 beats/min). From a balanced hypocaloric diet with a reduction of energy intake of 500 to 1000 kcal from the regular caloric intake, a total caloric intake reduction of 500 kcal/day was calculated. Such diets are suggested by the U.S. Food and Drug Administration (FDA) as the “standard therapy” for clinical trials [30].

### 2.2. Baseline and Postinterventional Testing

Medical history, anthropometric measurements, power performance tests, cycling spiroergometry, blood draws while fasting to gather laboratory data, and data collection during baseline and post-intervention testing after 8 weeks were all part of the data collection process. By applying the height measurement, the scale determined the body mass index (BMI). A simple stadiometer was used to measure height accurately (Seca 222, Seca Co., Birmingham, UK).

#### 2.2.1. Biochemical and Anthropometric Examination

At the beginning of this study, the BMIs of all subjects were computed. InBody Body Composition Analyzers were used to measure body composition (InBody230, InBody Co., Ltd., Seoul, Republic of Korea). Body weight and other measurements of body composition, such as lean body mass, body fat mass, BMI, water content, and body fat percentage, were taken, with measurements of the subject’s weight being taken to the nearest 0.5 kg.

Blood samples were taken before this study began and at its conclusion, eight weeks following the diet, for biochemical measurements. After centrifuging the serum, samples were kept in aliquots at 80 °C until additional analysis. The longest period of storage was 8 months.

Biochemical parameters were estimated by spectrophotometric methods using the ARCHITECT c Systems device (Abbott Park, IL, USA) in the Department of Biochemistry of the Institute for Clinical and Experimental Medicine in Prague. Serum levels of myokines (brain-derived neurotrophic factor (BDNF), fatty acid binding protein 3 (FABP3), fractalkine, follistatin-like protein 1 (FSTL1), IL6, musclin, oncostatin M, osteonectin) were analyzed by the multiplex assay MILLIPLEX^®^ Human Myokine Magnetic Bead Panel (HMYOMAG-56K; Merck KGaA, Darmstadt, Germany) and cytokines/chemokines (Interferon gamma (IFNγ), IL8, IL10, IL23, TNFα were measured by the multiplex assay MILLIPLEX^®^ Human Cytokine/Chemokine Magnetic Bead Panel (HCYTOMAG-60K; Merck KGaA, Darmstadt, Germany), both with using MAGPIX system (Luminex corporate, Austin, TX, USA). Serum levels of FGF21, FGF19, and C-reactive protein (CRP) were estimated by the highly sensitive ELISA kits (EH188RB, Invitrogen for FGF21; RD191107200R, BioVendor for FGF19; and BMS288INST, Invitrogen for CRP) using an Epoch microplate spectrophotometer (Agilent, Santa Clara, CA, USA).

#### 2.2.2. Strength and Aerobic Performance Testing

The training protocol was delineated in our primary paper [28]. In order to evaluate maximal power performance, the bench press, lat pull-down, and leg press were used as the three exercises in a strength performance evaluation. Aerobic performance testing was conducted by bicycle spiroergometry along with an analyzer of respiratory gases (Quark CPET, Cosmed, Concord, CA, USA). The participants were advised to work out until they became involuntarily exhausted and to keep their pedal cadence between 70 and 90 rpm. According to Gordon et al., we applied a modified exercise step regimen of 0.33 W·min^−1^. The subject’s inability to keep up a 40-rpm pedal cadence led the test to an end.

### 2.3. Diet Protocol

For eight weeks, subjects were randomly assigned, using an automated randomization mechanism, to either the CKD or the RD group. Prior to the start of the trial, subjects were required to attend an obligatory diet consultation with a nutritionist, who gave them thorough instructions on accurately recording their dietary food intake. The DietSystem software was used to enter and evaluate all food record data (DietSystem App, DietSystem App, s.r.o., Brno, Czech Republic). Detailed directions on what items would work for both types of diets were supplied to each participant. Additionally, according to randomization, participants undergo either an 8-week low-carb diet plan or a meal plan for a reduction diet. Each participant’s total daily calorie intake was lowered by 500 kcal based on their lifestyle (individually determined based on somatotype, physical activity, type of work, etc.).

Every week, a nutritionist checked on the participants’ overall diet compliance. Moreover, blood-hydroxybutyrate measurements at the completion of the research and twice-daily urinary ketone measures were used to assess adherence to CKD.

#### 2.3.1. Cyclical Ketogenic Reduction Diet

The CKD protocol included a five-day low-carbohydrate phase to induce and maintain ketosis, followed by a two-day carbohydrate phase (weekends) (nutrient ratio: carbs up to 30 g; proteins 1.6 g/kg; fats: computation of energy intake to substitute carbohydrates): (Intake of proteins 15%; carbs 8–10 g/1 kg of non–fat tissue 70%; and fat 15%).

#### 2.3.2. Nutritionally Balanced Reduction Diet

The RD diet protocol was established on the concepts of a healthy diet (nutrient ratio of carbohydrates 55%, fat 30%, and proteins 15% of total energy intake).

### 2.4. Training Protocol

The training protocol was delineated in our primary publication [28]. It consisted of a predefined combination of resistance training to support strength skills and aerobic training to increase endurance skills. A sport tester for aerobic performance and required check-in procedures at a gym were used to verify training compliance (TomTom Runner Cardio, TomTom, Amsterdam, The Netherlands).

### 2.5. Post-Intervention Testing

The methods used to gather data for the post-intervention testing were the same as those used for the baseline testing. The same researcher who conducted power measurements and performance testing for each subject at baseline did so to ensure reliability. Additionally, participants took their tests at the same time each day. After data analysis, test results were given to participants and compared to their baseline values.

### 2.6. Statistical Analysis

SigmaPlot 13.0 software (SPSS Inc., Chicago, IL, USA) was used for statistical analysis. Normally distributed data are listed as mean ± standard deviation (SD); non-parametric data are listed as median (interquartile range). The normality of all the data was estimated by the Shapiro–Wilk test. Intragroup differences were calculated by paired samples *t*-test or Wilcoxon signed rank test (non-parametric test). Two-way repeated measures ANOVA with Sidak’s post hoc test was utilized for testing intergroup differences. In all statistical tests, *p* values < 0.05 were deemed significant.

## 3. Results

### 3.1. The Influence of CKD vs. RD on Anthropometric, Biochemical and Hormonal Characteristics

Anthropometric parameters of subjects on CKD or RD at baseline (V1) and after 8 weeks of diet (V2) are delineated in Table 1. No significant differences between CKD and RD groups in anthropometric or body composition parameters were detected at baseline. Both CKD and RD lowered body weight, body fat mass, and body mass index (BMI), with similar effects of both diets. Lean body mass and body water content were significantly lowered by CKD, while they were not influenced by RD (Table 1).

Biochemical and hormonal parameters of study subjects with CKD or RD at V1 and V2 are shown in Table 2. No differences between CKD and RD groups in biochemical and hormonal parameters were found at V1. CKD significantly increased total, LDL, and HDL cholesterol as well as β-OH-butyrate concentrations. CKD also decreases blood glucose levels. In the RD group, total cholesterol and uric acid concentrations decreased significantly after the intervention, while other parameters did not change. Total and LDL cholesterol levels at the end of the intervention (V2) were significantly lower in the RD vs. CKD group.

### 3.2. The Influence of CKD vs. RD on Serum Concentrations of Myokines

Serum concentrations of selected myokines, adipokines, and cytokines in subjects on CKD or RD at baseline (V1) and after 8 weeks of diet (V2) are shown in Table 3. No differences in serum myokine concentrations between the CKD and RD groups were found at V1. At V2, serum osteonectin levels were significantly lower in the RD group as compared to the CKD group.

In the RD group, serum osteonectin concentrations significantly decreased after the intervention, while no change in the concentrations of this myokine was found in the CKD group. Increased levels of musclin were found at V2 in subjects from the RD group compared to V1, while no such changes were detected in the CKD group. In contrast, the FGF21 level significantly decreased at V2 in the CKD group while it remained unchanged in the RD group. No significant changes in the rest of the parameters were detected (Table 3).

## 4. Discussion

The interactions between diet and physical activity play a major role in the long-term regulation of body weight and physical fitness. Endocrine factors, including not only classical hormones but also adipokines, cytokines, and myokines, were shown to be very active players in this process [10,31,32]. The aim of our study was to comprehensively analyze the changes in body composition, physical fitness, and serum concentrations of selected myokines and cytokines in healthy young males undergoing combined resistance/aerobic training in combination with two dietary protocols: CKD or RD.

This study follows up on our earlier findings [28], where eight weeks of regular exercise training combined with either CKD or RD decreased body weight and body fat to a similar degree, while muscle strength and endurance performance were improved only in the RD group. These changes were seen on the background of slightly reduced lean body mass and body water content. It was significantly reduced in CKD subjects, while RD had no effect on it. The large decline in body water in the CKD group was most likely caused by low carbohydrate intake-related glycogen depletion.

Myokines, muscle-secreted molecules, play a crucial role in the regulation of muscle metabolism and function. Out of the myokine/adipokine/cytokine panel analyzed, the serum concentrations of three myokines were affected by the intervention. Musclin is a peptide produced by bone and skeletal muscle cells that plays an important role in the regulation of bone growth and physical endurance by enhancing the growth of muscle fibers [11,12,13]. Given that musclin has been demonstrated to contribute to muscle growth and regeneration [33], it is tempting to speculate that its increased concentration in the RD group could have contributed to the increased lean body mass and improved exercise performance detected in the RD group. In support of this hypothesis, musclin levels were unchanged in the CKD group, where no changes in muscle strength or endurance performance were detected. Recently, it has been reported that runners with sufficient carbohydrate intake have higher levels of musclin, which may contribute to improved adaptations to exercise, such as improved glucose homeostasis and browning of adipose tissue [34]. These findings are again in line with the increased musclin levels and enhanced endurance performance observed in our RD subjects.

Osteonectin is an acidic extracellular matrix glycoprotein that plays an important role in bone mineralization and collagen binding [35]. In our study, serum osteonectin levels decreased in the RD group while they remained unaffected in the CKD group. The interconnection between osteonectin, exercise, and metabolic and energy homeostasis regulation is still not clear. In experimental studies, osteonectin deficiency was accompanied by an accelerated aging phenotype and reduced physical activity [36]. These changes were attributed to reduced extracellular matrix mass and decreased collagen maturity. In other studies, osteonectin has also been described as a metabolic enhancer of skeletal muscle growth, with its mRNA expression being induced by exercise in both mice and humans [37]. The reason for decreased osteonectin levels after 8 weeks of exercise in our study is thus unclear. One possible explanation is that most of the studies published so far were performed in rodents, which may not be representative of the regulation of osteonectin levels in humans. Furthermore, in the human study by Aoi and colleagues [37], osteonectin levels were increased only when measured shortly after exercise and returned to baseline values six hours after exercise. Therefore, as our study measured osteonectin levels in the fasting state, we would not be able to detect any post-exercise changes in its levels.

FGF21 is a novel metabolic regulator produced primarily by the liver, but to a certain degree also by other tissues such as white and brown adipose tissue, skeletal muscle, and the pancreas [38]. FGF21 has potent antidiabetic and lipid-lowering effects in animal models of obesity and type 2 diabetes mellitus. This hormone contributes to body weight regulation and is strongly involved in the response to nutritional deprivation and the ketogenic state in mice [39]. A systematic review of clinical studies exploring the effect of exercise on FGF21 levels concluded that acute exercise tended to increase circulatory levels of FGF21, while chronic exercise with a duration over 4 weeks had rather the opposite effect [40]. It was also described that higher levels of daily physical activity could decrease circulating FGF21 levels [41]. Moreover, it has been reported that carbohydrate intake correlates with levels of FGF21 [34]. In our study, FGF21 levels decreased in the CKD group but remained unchanged in the RD group, suggesting that the effects of the ketogenic diet could have interacted with exercise.

In addition to the important role of myokines in the regulation of muscle regeneration and physical performance, these molecules may also represent a promising therapeutic target for metabolic diseases [42]. A sedentary lifestyle without a low level of physical activity is then directly proportional to the development of many chronic diseases accompanied by systemic inflammation [43,44]. Increased glucose levels in obese patients with type 2 diabetes can disrupt the regulation of vascular tension and, among other things, humoral and cellular inflammatory processes [45]. Several studies have confirmed the role of ketogenic diet consumption on body weight reduction, body composition change, and regulation of cytokine and adiponectin levels [46,47,48,49,50]. The lack of significant changes in pro-inflammatory cytokines and adipokines in the CKD group in our study as compared to its decrease in patients with obesity and type 2 diabetes after the ketogenic diet may be due to the fact that these positive effects are seen in the context of metabolic impairments and subclinical inflammation but not in metabolically healthy lean subjects. Taken together, the ketogenic diet and nutritionally balanced reduction diet are two different approaches to weight loss and overall health improvement, with some overlap in their benefits but also differences in the advantages and disadvantages of each approach (see Table 4 for the summary).

For proper interpretation of the results of our study within the context of other published data, it is important to consider its strengths and limitations. The randomized design and the good compliance of the subjects to the dietary and treatment regimens are strong points of our trial. On the other hand, the limitations include a relatively short duration, a low number of subjects, and the inclusion of only male participants. Another limitation may be the use of bioelectrical impedance analysis instead of dual-energy X-ray absorptiometry, which is more accurate.

## 5. Conclusions

Myokines play a crucial role in the communication between skeletal muscles and other distinct organs in order to adapt whole-body metabolism to nutritional changes. Our data suggest that changes in systemic osteonectin and musclin levels could contribute to improved muscle strength and endurance performance in healthy young males on RD undergoing regular resistance/aerobic training as compared to subjects on CKD, where no improvements in muscle strength and endurance performance were detected. Further studies with a longer duration and on a higher number of subjects are warranted to support the validity of these findings for other populations, such as professional athletes or patients with obesity and metabolic complications.

## Figures and Tables

**Table 1 nutrients-15-01720-t001:** Anthropometric parameters of study subjects with CKD or RD at baseline (V1) and after 8 weeks of diet (V2).

Parameters	CKD		RD	
	V1	V2	V1	V2
Age (year)	23 ± 5		24 ± 4	
Height (cm)	181 ± 6		186 ± 10	
BMI (kg/m^2^)	26.1 ± 3.7	24.6 ± 3.3 *	26.9 ± 4.3	25.5 ± 4.2 *
WEIGHT (kg)	85.6 ± 13.4	81.0 ± 12.0 *	93.0 ± 17.5	88.5 ± 17.4 *
MUSCLES (kg)	41.8 ± 4.5	40.0 ± 4.6 *	43.5 ± 5.3	43.1 ± 5.3
FAT (kg)	12.9 ± 6.9	11.0 ± 5.8 *	17.6 ± 9.8	13.6 ± 9.0 *
% FAT	14.5 ± 5.5	13.0 ± 5.1 *	17.9 ± 6.9	14.2 ± 6.9 *
WATER (kg)	53.2 ± 5.6	51.0 ± 5.6 *	55.1 ± 6.4	54.8 ± 6.5

Data are shown as mean ± SD; statistical significance of intragroup differences is from paired samples *t*-test or Wilcoxon signed rank test (V1—baseline testing vs. V2—testing after 8 weeks of diet). * *p* < 0.05 V1 vs. V2. BMI: Body mass index.

**Table 2 nutrients-15-01720-t002:** Biochemical and hormonal parameters of study subjects with CKD or RD at baseline (V1) and after 8 weeks of diet (V2).

Parameters	CKD		RD	
	V1	V2	V1	V1
Total cholesterol (mmol/L)	3.92 ± 0.52	4.92 ± 0.74 *	4.47 ± 0.67	4.03 ± 0.80 *^Δ^
Triglycerides (mmol/L)	0.89 (0.65–1.11)	0.8 (0.66–0.93)	0.92 (0.71–1.03)	0.81 (0.66–1.01)
LDL cholesterol (mmol/L)	2.37 ± 0.41	3.14 ± 0.64 *	2.73 ± 0.52	2.43 ± 0.63 ^Δ^
HDL cholesterol (mmol/L)	1.15 ± 0.24	1.39 ± 0.28 *	1.30 ± 0.39	1.20 ± 0.30
Fasting glucose (mmol/L)	5.28 ± 0.34	4.96 ± 0.47 *	5.26 ± 0.36	5.22 ± 0.36
Insulin (µIU/mL)	5.37 (3.99–8.29)	5.86 (3.46–7.42)	6.35 (3.13–13.34)	8.44 (4.76–10.98)
Uric acid (mmol/L)	357 (312.5–430.5)	350 (324.5–421.5)	397 ± 63	368 ± 53 *
CK (ukat/L)	4.40 ± 2.81	2.81 ± 1.21	3.80 ± 2.03	3.03 ± 2.03
LDH (ukat/L)	2.68 ± 0.60	2.47 ± 0.42	2.74 ± 0.44	2.55 ± 0.33
β-OH-butyrate (mmol/L)	0.1 (0.1–0.2)	0.2 (0.1–0.6) *	0.1 (0.1–0.3)	0.1 (0.1–0.2)

Normally distributed data are shown as mean ± SD; non-parametric data are expressed as median (interquartile range). Statistical significance of intragroup differences is from paired samples *t*-test or Wilcoxon signed rank test (V1—baseline testing vs. V2—testing after 8 weeks of diet). * *p* < 0.05 V1 vs. V2; for intergroup differences, significance is from a Two-way repeated measures ANOVA with Sidak’s post hoc test. ^Δ^
*p* < 0.05 V2 CKD vs. V2 RD. LDL cholesterol: low-density lipoprotein cholesterol; HDL cholesterol: high-density lipoprotein cholesterol; CK: Creatine kinase; LDH: Lactate dehydrogenase; β-OH-butyrate—β-hydroxy-butyrate.

**Table 3 nutrients-15-01720-t003:** Changes in serum myokines, adipokines, and cytokines of study subjects on CKD or RD at baseline (V1) and after 8 weeks of diet (V2).

Parameters	CKD		RD	
	V1	V2	V1	V2
Oncostatin M (pg/mL)	8.26 (5.16–10.6)	8.75 (5.75–11.2)	10.5 (8.63–12.5)	10.9 (9.35–17.9)
Musclin (pg/mL)	48.6 (26.8–80)	55.8 (36.5–83.2)	32.9 (12.2–85.8)	74.5 (34.7–95.4) *
Osteonectin (pg/mL)	630 (489–701)	596 (529–803)	562 (490–665)	511 (484–568) *^∆^
BDNF (ng/mL)	11.9 (10.9–13.3)	12.9 (10.5–13.5)	11.6 (10.1–13.5)	11.9 (11.2–13)
FABP3 (ng/mL)	1.01 (0.87–1.34)	1.06 (0.87–1.55)	1.27 (0.79–1.98)	1.17 (0.88–1.97)
FSTL1 (ng/mL)	2.75 (1.1–4.89)	2.95 (2.07–4.84)	3.64 (1.58–7.09)	3.79 (1.63–7.29)
FGF19 (pg/mL)	194 (134–327)	207 (119–292)	165 (120–210)	133 (120–222)
CRP (mg/L)	1.02 (0.3–2.5)	0.85 (0.19–2.34)	0.69 (0.28–1.6)	0.71 (0.16–1.29)
FGF21 (pg/mL)	181 (112–709)	86.4 (45.1–571) *	272 (176–1138)	193 (144–1142)
Fractalkin (pg/mL)	241 (213–315)	208 (183–316)	211 (191–247)	222 (202–300)
IFNγ (pg/mL)	17.6 ± 8.2	16.5 ± 8.0	16.3 ± 5.1	17.5 ± 5.1
IL10 (pg/mL)	11.5 ± 6.1	10.5 ± 6.5	12.1 ± 8.4	11.6 ± 7.2
IL23 (pg/mL)	265 ± 134	246 ± 126	272 ± 107	284 ± 122
IL6 (pg/mL)	3.46 (1.16–5.82)	2.72 (1.02–4.13)	2.13 (1.01–4.24)	2.68 (1.44–5.52)
IL8 (pg/mL)	9.8 (8–20.6)	11.4 (7.5–18.4)	9.7 (7.4–11.1)	11.2 (8.3–22.3)
TNFα (pg/mL)	8.85 (6.93–12.19)	9.14 (7.32–12.74)	9.07 (7.46–10.41)	11.38 (6.94–16.56)

Normally distributed data are shown as mean ± SD; non-parametric data are expressed as median (interquartile range). Statistical significance of intragroup differences is from paired samples *t*-test or Wilcoxon signed rank test (V1—baseline testing vs. V2—testing after 8 weeks of diet). * *p* < 0.05 V1 vs. V2; for intergroup differences, significance is from a Two-way repeated measures ANOVA with Sidak’s post hoc test. ^Δ^
*p* < 0.05 V2 CKD vs. V2 RD. BDNF: Brain-derived neurotrophic factor; FABP3: Fatty acid binding protein 3; FSTL1: Follistatin-related protein 1; FGF19: Fibroblast growth factor 19; CRP: c-reactive protein; FGF21: Fibroblast growth factor 21; IFNγ: Interferon gamma; IL10/23/6/8: Interleukin 10/23/6/8; TNFα: tumor necrosis factor α.

**Table 4 nutrients-15-01720-t004:** Comparison of advantages and disadvantages of the CKD and RD.

CKD		RD	
Advantages	Disadvantages	Advantages	Disadvantages
↓ fasting glucose	↔ VO_2max_	sustainable weight loss	slow initial weight loss
↓ weight	↔ TTE	↓ adipose tissue	feel hunger and cravings
↓ adipose tissue	↓ strength	↑ strength	
feel satiety	↓ LBM	↑ endurance	
↑ cognitive function	↑ keto flu	all nutrients	
	↑ LDL cholesterol	variation of food	
	↓ fiber	↑ adherence	
	nutrient deficiencies		
	↓ adherence		
	↑ depression		

TTE: time to exhaustion; LBM: lean body mass; LDL cholesterol: low density cholesterol; VO_2_max: peak oxygen uptake. ↓ decrease, ↑ increase, ↔ unchanged.

## Data Availability

The data presented in this study are available upon request from the corresponding author.

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
