# Peer review of "Different Effects of Cyclical Ketogenic vs. Nutritionally Balanced Reduction Diet on Serum Concentrations of Myokines in Healthy Young Males Undergoing Combined Resistance/Aerobic Training"

_nutrients, 2023, doi:10.3390/nu15071720_

Round 1

Reviewer 1 Report

This study aimed to explore the effects of cyclical ketogenic reduction diet (CKD) vs. nutritionally balanced reduction diet (RD) combined with regular resistance/aerobic training in healthy young males on serum concentrations of myokines and their potential role in changes of physical fitness. I suggest few changes prior the acceptance for publication.

-Abstract: to add more "numbers"in the results section.

-Introduction: The authors did not investigate athletes, thus, the introduction must be rewritten. 

What is hypothesis of study? Moreover, in the end of introduction is adequate to add the main aim of study.

-Methods: To add in more details the design of study. The authors cited the reference 22 as previous study, however, the description of methods is shorter and difficult of read of manuscript. Has outlier values in blood variables? 

-results: Table 1, to remover the line "number (n)".

-Discussion: in the second paragraph as well as all discussion is not clear the changes of water content in the CKD group. This must her discussed.

In addition, what is limitation of study? the use of BIA, but not DXA? or US?  What is food intake analyses?

In general, these data must be analyzed using the ANOVA two way.

Reviewer 2 Report

This manuscript must be drastically revised for multiple reasons. Please see below my suggestions:

I have never saw 7 pages of real manuscript with 16!!!! authors, following 25 patients (less than 2/author). Please read the Academic ethic in publication.

L58. Develop the effects/role of ketogenic diet (and of other diets, in general) in different diseases/disorders – I suggest checking and referring to very recent publications I found fast relevant in the topic as  https://doi.org/10.1007/s12035-020-02065-3  https://doi.org/10.1016/j.lfs.2020.118661 https://doi.org/10.1007/s11356-021-17589-x  https://doi.org/10.3390/molecules27206804 and doi: 10.4183/aeb.2020.470  . Detailing this idea will also improve the reason for choosing this topic 

Aim of the study. L101-107. Please make it a separate paragraph of Introduction. Highlight the novelty your study brings to the field. Remove the part describing what was done through the research, as it is a repetitive information with the following section. Make a relevant aim of the study.

Subsection 2.1The number of cases included in your article is too low in order to obtain statistical signification. I am sure that the authors understand/know that the no. of 25 subjects is much too small for a relevant statistic. To have a statistically significant statistic (comparative or not), each group must have at least 33 individuals. Extend your study or explain properly. Redesign this part as in this shape it is not enough for any publication.

L124. Declaration of Helsinki must be referenced as follows: WMA Declaration of Helsinki - Ethical Principles for Medical Research Involving Human Subjects Available online: https://www.wma.net/policies-post/wma-declaration-of-helsinki-ethical-principles-for-medical-research-involving-human-subjects/  (accessed on date)

Abbreviations used in the tables must be explained under the table.

Be constant with denotation. Table 3. ml must be written as mL, as Litter is the international unit of measure for volume. Revise the entire manuscript in this regard.

The Discussion chapter should be improved. Please discuss in a separate Table previous studies that addressed the same thematic/topic and their results. Some of the references I suggested in the Introduction can be consulted for information in this regard.

Please make a figure where you present advantages/disadvantages of the two types of diets in an easy to read/follow form. 

After L349, please add a new paragraph, highlighting the strengths and the weakness (number of patients/subjects is for sure a big weakness) of your paper/results.

Round 2

Reviewer 1 Report

no more comments

Author Response

We thank you very much for your previous suggestions and comments.

Reviewer 2 Report

L134. - Ethical Principles for Medical Research Involving Human Subjects available online: https://www.wma.net/policies-post/wma-declaration-of-helsinki-ethical-principles-for-medical-research-involving-human-subjects/ (accessed on January 2nd, 2013) must be corrected the DATE and inserted as Reference (in the brackets, as number) and the entire text to be found in the reference section.

For the Tables 1-3, please complete the head of the table for the 1st column (parameters, characteristics, or as the authors consider).
